# Optimization of a Protocol for Protein Extraction from Calcified Aortic Valves for Proteomics Applications: Development of a Standard Operating Procedure

**DOI:** 10.3390/proteomes10030030

**Published:** 2022-09-01

**Authors:** Fábio Trindade, Ana F. Ferreira, Francisca Saraiva, Diana Martins, Vera M. Mendes, Carla Sousa, Cristina Gavina, Adelino Leite-Moreira, Bruno Manadas, Inês Falcão-Pires, Rui Vitorino

**Affiliations:** 1Cardiovascular R&D Centre—UnIC@RISE, Department of Surgery and Physiology, Faculty of Medicine, University of Porto, 4200-319 Porto, Portugal; 2CNC—Center for Neuroscience and Cell Biology, University of Coimbra, 3004-517 Coimbra, Portugal; 3Cardiology Department, Centro Hospitalar Universitário de São João, 4200-319 Porto, Portugal; 4Cardiology Department, Hospital Pedro Hispano, Unidade Local de Saúde de Matosinhos, 4464-513 Matosinhos, Portugal; 5Department of Cardiothoracic Surgery, Centro Hospitalar Universitário de São João, 4200-319 Porto, Portugal; 6Institute for Interdisciplinary Research, University of Coimbra, 3030-789 Coimbra, Portugal; 7iBiMED—Institute of Biomedicine, Department of Medical Sciences, University of Aveiro, 3810-193 Aveiro, Portugal; 8LAQV/REQUIMTE, Department of Chemistry, University of Aveiro, 3810-193 Aveiro, Portugal

**Keywords:** aortic valve, calcification, homogenization, protein extraction, zirconium dioxide beads, proteomics, SOP

## Abstract

The comprehension of the pathophysiological mechanisms, the identification of druggable targets, and putative biomarkers for aortic valve stenosis can be pursued through holistic approaches such as proteomics. However, tissue homogenization and protein extraction are made difficult by tissue calcification. The reproducibility of proteome studies is key in clinical translation of the findings. Thus, we aimed to optimize a protocol for aortic valve homogenization and protein extraction and to develop a standard operating procedure (SOP), which researchers can use to maximize protein yield while reducing inter-laboratory variability. We have compared the protein yield between conventional tissue grinding in nitrogen followed by homogenization with a Potter apparatus with a more advanced bead-beating system. Once we confirmed the superiority of the latter, we further optimized it by testing the effect of beads size, the number of homogenization cycles, tube capacity, lysis buffer/tissue mass ratio, and two different lysis buffers. Optimal protein extraction was achieved with 2.8 mm zirconium dioxide beads, in two homogenization cycles, in the presence of 20 µL RIPA buffer/mg tissue, using 2 mL O-ring cryotubes. As a proof of concept of the usefulness of this SOP for proteomics, the AV proteome of men and women with aortic stenosis was characterized, resulting in the quantification of proteins across six orders of magnitude and uncovering some putative proteins dysregulated by sex.

## 1. Introduction

Degenerative aortic valve stenosis (AS) remains the main valvular heart disease requiring surgery or transcatheter intervention in Europe [1]. So far, no pharmacological treatment has shown efficacy to revert or at least stall the progression of AS in clinical trials [2,3,4]. Despite the progress in recent decades, in our understanding of AS pathophysiology, we still do not hold biomarkers for an earlier diagnosis of this silent condition, and aortic valve (AV) replacement continues to be the only effective treatment available. Proteomics of human AVs has been a strategy used to clarify the mechanisms underlying the fibrocalcific transformation of the AV [5,6,7,8,9,10,11,12,13], which may translate into new druggable targets and unveil potential new biomarkers [14,15,16,17].

Apart from a well-framed question, study design and cohort, we have to consider analytical variance to obtain a meaningful proteome dataset with potential for clinical translation of the findings. We cannot fully control biological variability when working with human samples, such as aortic valves. Even after carefully defining eligibility criteria and applying powerful matching strategies, there are always uncontrolled sources of proteome variation. These may encompass the timing of collection (circadian effects), fasting, hormonal status, seasonal effects, comorbidities, and unpredicted interferences due to medications. Therefore, analytical variance should be minimized at least, and this can be accomplished by developing and implementing standard operating procedures (SOPs) for sample collection and processing [18,19].

Herein, we propose a protocol to homogenize AVs and extract proteins by applying a state-of-the-art bead-beating system to maximize protein recovery while reducing sample manipulation. We demonstrate its usefulness by profiling the proteome of six human AVs by label-free Sequential Window Acquisition of All Theoretical Mass Spectra (SWATH-MS). We also share an optimized standard operating procedure (SOP) to increase the reproducibility of AV proteomics studies, facilitating the clinical translation of the findings.

## 2. Material and Methods

### 2.1. Sample Collection

Eleven AVs were sourced from our local biobank. These samples have been collected on the occasion of isolated surgical aortic valve replacement or concomitant to other surgical procedures. This study followed the principles stated in the Declaration of Helsinki, and the ethics committee of the Centro Hospitalar Universitário de São João approved the protocol (reference CEC109-2020). Informed consent was obtained from all patients. After surgical resection, the samples were transferred into a preservation solution (Custodiol-CE Bretschneider HTK solution, Dr. Franz Köhler Chemie Pharmaceuticals GmBH), kept at 4 °C, and transported to the lab within 2 h of collection. Then, the samples were washed with phosphate-buffered saline (PBS) to remove any remaining blood contaminants and flash-frozen in liquid nitrogen. The samples were stored at −80 °C, until further processing.

### 2.2. Tissue Homogenization and Protein Extraction Optimization

We tested several conditions to aim for the highest protein extraction yield and the most representative protein electrophoretic profile possible. We compared liquid nitrogen grinding followed by mechanical disruption with a Potter–Elvehjem apparatus (Glas-Col, Terre Haute, IN, USA) with a bead-beating system (P000673-MLYS0-A, Minilys, Bertin Instruments, Montigny-le-Bretonneux, France) using zirconium dioxide beads (Bertin Technologies, Montigny-le-Bretonneux, France). We further optimized the latter protocol by testing the effect of beads’ size (1.4 mm vs. 2.8 mm), tube capacity (2 vs. 7 mL, Sarstedt, Nümbrecht, Germany), extraction cycles (one up to four), lysis buffer/tissue mass ratios (10, 15 and 20 µL/mg tissue) and lysis buffer type (RIPA vs. urea). The RIPA lysis buffer was prepared based on a commercial formulation (25 mM Tris-HCl pH 7.6, 150 mM NaCl, 1% NP-40, 1% sodium deoxycholate, 0.1% SDS; 89900, Thermo Fisher Scientific, Waltham, MA, USA), enriched with EDTA to a final concentration of 1 mM (EDTA disodium salt dihydrate, ≥99% Merck-Millipore, Darmstadt, Germany), a protease inhibitor cocktail (Halt PIC, EDTA-free 100×, Thermo Fisher Scientific), and a phosphatase inhibitor cocktail (PhosSTOP, Roche, Mannheim, Germany). The urea buffer (7 M urea, >99.5% Acros Organics, 2 M thiourea, AppliChem A2535, 1% SDS, >99% Fisher BioReagents, 1 mM dithiothreitol, Nzytech, Lisbon, Portugal) was prepared in house and also supplemented with 1 mM EDTA, and the same protease and phosphatase inhibitor cocktails. In any case, the protein lysates were centrifuged at 13,680× *g*, for 15 min at 4 °C (Mikro 200 R centrifuge, Hettich Zentrifugen, Tuttlingen, Germany), and the protein-rich supernatants stored at −80 °C until further processing. The protocol details and variations are explained in the “Results” section and the optimized SOP is provided as Appendix A.

### 2.3. Protein Quantification

Protein concentration was estimated using the detergent-compatible kit (DC kit) from Bio-Rad Laboratories, Hercules, CA, USA. When comparing the performance of the RIPA lysis buffer with the urea lysis buffer, the samples were pre-precipitated with acetone overnight at −20 °C, due to the method’s incompatibility with large amounts of urea. All samples (including those processed with RIPA buffer) were then centrifuged at 12,000× *g* for 30 min, at 4 °C, and solubilized in 1% SDS. In any case, bovine serum albumin was used as the standard.

### 2.4. SDS-PAGE

Protein extracts were separated by SDS-PAGE gels following Laemmli procedure [20]. Essentially, 25 µg of protein from each sample was diluted to the same concentration and mixed with Laemmli buffer, incubated for 5 min at 95 °C and loaded into hand-cast 12% Tris-HCl gels. The proteins were separated under reducing and denaturing conditions using the Bio-Rad SDS-PAGE system, for 20 min applying a 120 V voltage, and then for 30 min applying a 200 V voltage. The gels were then incubated in a fixation solution (40% methanol, 10% acetic acid) for 30 min, stained with 0.12% Colloidal Coomassie Blue G250 in 20% methanol overnight, and destained with 20% methanol until an optimal contrast was achieved. Gels were scanned with ChemiDoc system and analyzed with ImageLab (version 6.1, BioRad Laboratories, Hercules, CA, USA) for automatic detection of protein bands.

### 2.5. Proteomics Analysis

#### 2.5.1. Sample Preparation

Each sample was mixed with 6× concentrated Laemmli buffer with bromophenol blue to a final concentration of 5% glycerol, 1.7% SDS, 100 mM dithiothreitol, and 50 mM Tris buffer at pH 6.8. Proteins were denatured and reduced by boiling for 5 min at 95 °C and alkylated with a 40% acrylamide solution (1:15 *v*/*v*). A Short-GeLC approach [21] was followed, where the proteins were separated by SDS-PAGE for ~19 min at 110 V and stained with Coomassie Brilliant Blue G-250. For protein identification by data-dependent acquisition (DDA) two pools of samples were prepared (male and female) where 60 μg of protein were loaded into the gel and each lane was divided into 5 fractions. For SWATH experiments [a data-independent acquisition (DIA) approach], 40 μg of protein of each sample were loaded individually and each lane was divided into three fractions. Gel bands were destained using a 50 mM ammonium bicarbonate with 30% acetonitrile solution, followed by overnight protein digestion with trypsin, considering an enzyme:protein ratio of 1:20 (*w*/*w*). Peptide extraction was performed by incubating the gel pieces with solutions containing increasing percentages of acetonitrile (30, 50, and 98%) with 1% of formic acid. For protein identification, the ten pool fractions were analyzed separately by DDA acquisition and for protein quantification, the three fractions from each sample were joined and a single analysis per sample was performed by SWATH-MS (DIA acquisition).

#### 2.5.2. LC-MS Methodology

Samples were analyzed on a NanoLC™ 425 System (Eksigent, Dublin, OH, USA) coupled to a Triple TOF™ 6600 mass spectrometer (Sciex, Framingham, MA, USA) and the ionization source (ESI DuoSpray™ Source from Sciex, Framingham, MA, USA). Peptides were separated on a Triart C18 Capillary Column 1/32” (12 nm, 3 μm, 150 mm × 0.3 mm, YMC, Dinslaken, Germany) and using a Triart C18 Capillary Guard Column (0.5 mm × 5 mm, 3 μm, 12 nm, YMC, Dinslaken, Germany) at 50 °C. The flow rate was set to 5 μL/min and mobile phases A and B were 5% DMSO plus 0.1% formic acid in water and 5% DMSO plus 0.1% formic acid in acetonitrile, respectively. The LC program was performed as follows: 5–30% of B (0–50 min), 30–98% of B (50–52 min), 98% of B (52–54 min), 98–5% of B (54–56 min), and 5% of B (56–65 min). The ionization source was operated in the positive mode set to an ion spray voltage of 5500 V, 25 psi for nebulizer gas 1 (GS1), 10 psi for nebulizer gas 2 (GS2), 25 psi for the curtain gas (CUR), and source temperature (TEM) at 100 °C. For DDA experiments, the mass spectrometer was set to scan full spectra (*m*/*z* 350–2250) for 250 ms, followed by up to 100 MS/MS scans (*m*/*z* 100–1500). Candidate ions with a charge state between + 1 and + 5 and counts above a minimum threshold of 10 counts per second were isolated for fragmentation, and one MS/MS spectrum was collected before adding those ions to the exclusion list for 15 s (mass spectrometer operated by Analyst^®^ TF 1.8.1, Sciex^®^). The rolling collision energy was used with a collision energy spread of 5. For SWATH experiments, the mass spectrometer was operated in a looped product ion mode and specifically tuned to a set of 42 overlapping windows, covering the precursor mass range of 350–1400 *m*/*z*. A 50 ms survey scan (*m*/*z* 350–2250) was acquired at the beginning of each cycle, and SWATH-MS/MS spectra were collected from 100 to 2250 *m*/*z* for 50 ms, resulting in a cycle time of 2.2 s.

#### 2.5.3. Data Analysis—Ion-Library Construction (DDA)

An ion library of the precursor masses and fragment ions was created by combining all files from the pools (5 gel fractions × 2 pools) in a single protein identification search using the ProteinPilot™ software (v5.0, Sciex, Framingham, MA, USA). The paragon method parameters included a search against the reviewed Human (SwissProt) database (20,375 sequences downloaded on 9 February 2022), cysteine alkylation by acrylamide, digestion by trypsin, and gel-based ID. An independent False Discovery Rate (FDR) analysis was performed using the target-decoy approach provided by Protein Pilot™, to assess the quality of identifications.

#### 2.5.4. Data Analysis—Relative Quantification of Proteins by SWATH-MS

SWATH data processing was performed using the SWATH™ plug-in for PeakView™ (v2.0.01, Sciex^®^, Framingham, MA, USA). Protein relative quantification was performed in all samples using the information from the protein identification search. Quantification results were obtained for peptides quantified with less than 1% of FDR in at least one of the samples by performing the sum of up to 5 fragments/peptide. Peptide relative peak areas were normalized for the total sum of areas for the respective sample, and protein quantities were obtained by the sum of up to 15 peptides/protein.

The MS data have been deposited to the ProteomeXchange Consortium via the PRIDE partner repository with the dataset identifier PXD034386.

#### 2.5.5. Bioinformatics Analyses

Significantly dysregulated proteins were subjected to protein–protein interaction analysis with STRING webtool (version 11.5, https://string-db.org/, accessed on 16 May 2022). A score of 0.4 was set as the minimum threshold for consideration of the validated and putative interactions.

### 2.6. Statistical Analysis

Apart from protein identification/quantification, which was performed with the ProteinPilot and PeakView software (Sciex Framingham, MA, USA), all statistical analyses were performed using R statistical built-in functions and the package ‘matrixTests’ (Karolis Koncevicius, CRAN repository).

Data are expressed as the mean ± standard deviation, except for the protein yield graphs, representing the median and interquartile range.

Normality was inquired with Shapiro’s test. To detect differences in protein yields across the various AV homogenization conditions tested, we used an unpaired two-tailed t-test or a one-way ANOVA followed by a post hoc Tukey’s honest significance test, depending on whether two or three conditions were being tested.

Dysregulated proteins were assessed by calculating the ratio (fold change) between proteins in women and men. Welch’s corrected unpaired two-tailed t-test was used to filter significant differences. Only proteins quantified with at least 2 peptides were considered.

In any case, a *p*-value lower than 0.05 was considered significant.

## 3. Results and Discussion

In conditions such as aortic stenosis, the aortic valve suffers extensive fibrosis and calcification, becoming stiff and hard, and acquiring bone-like properties. In these cases, protein extraction can be very challenging. Therefore, we hypothesized that protein extraction would be facilitated by the use of rigid beads, such as zirconium dioxide, in a bead-beating system. Thus, we started to compare the protein extraction yield using a bead-beating system (BBS) with a traditional liquid nitrogen grinding followed by mechanical disruption with a Potter–Elvehjem pestle in a glass tube (LN-PEP), already in use in our lab.

### 3.1. The Bead-Beating System Outperforms the Traditional Liquid Nitrogen Grinding Followed by Mechanical Disruption with a Potter–Elvehjem Pestle

We started by selecting three random AV tissue samples to be processed by the two protocols (LN-PEP vs. BBS). The initial tissue mass was recorded (53 ± 4 mg). For the traditional LN-PEP method, each sample was transferred to a pre-chilled mortar and grounded with nitrogen with the help of a ceramic pestle. The sample fragments were transferred to a 1.5 mL tube and the final tissue mass was recorded. On average, 31% of the initial tissue was lost during this process. Then, 10 µL of RIPA buffer was added for every mg of tissue and the samples vortexed for 30 s. To further homogenize the tissue and extract the protein, the samples were transferred to a glass tube and subjected to 10 + 10 Potter strokes at a velocity of 30×, followed by 10 + 10 strokes at a velocity of 50×, always interspersed by 30 s on ice. The final lysates were recovered to a new tube and centrifuged to clear the supernatants. After estimating protein concentration, we observed a final LN-PEP’s protein yield of 1.3 ± 0.3% (m protein/m tissue). For the BBS, each sample was grossly fragmented (not grounded) in a pre-chilled mortar and transferred to an O-ring cryotube (initial tissue mass of 56 ± 17 mg). Next, the zirconium dioxide beads and the same RIPA buffer, at the same ratio (10 µL/mg tissue), were added to the tubes and these were tightly screwed. Tissue homogenization was promoted by subjecting the samples to one or two 5000 rpm beating cycles for 30 s, interspersed by a 5 min rest on ice. After a 2 min spin down at 2000 rpm, the resulting lysates were transferred to a new tube and centrifuged in the same conditions to obtain the protein-rich supernatants. The many variations and details of the BBS protocol aiming for the best protein yield are explained in the following sections. At this point, we observed no tissue loss when AVs were homogenized by the BBS method. This is because the sample does not need to be fully grounded and because a complete homogenization takes place inside screw-capped tubes, which provide a closed system for tissue disruption. The BBS was more efficient than the conventional LN-PEP method, achieving a mean protein yield of 1.5 ± 0.7%, though not significant. We also compared the SDS-PAGE profile of the attained protein extracts (Figure 1), and despite an overall similar lane profile for each sample when comparing LN-PEP with BBS, the number of bands (automatically detected) was increased when using the beads, which suggests a better proteome coverage. For all these reasons, we attribute a superior performance of the beads in extracting protein from calcified AV tissue. We then aimed to improve the protein yield by optimizing the protocol in many different aspects, as follows.

### 3.2. The Effect of Increasing Beads Size and a Second Homogenization Cycle

Aiming at the best protein yield possible, we tested different conditions in the BBS method, starting by changing the size of the beads and the number of homogenization cycles. All experiments were conducted with replicates of three different AV samples. Based on the protocol described in the previous section, we have compared the efficiency of 1.4 mm (medium-sized) versus 2.8 mm (large) zirconium dioxide beads, and one versus two beating cycles (Figure 2—of note, one sample was lost before quantification, explaining the total of 11 samples depicted in the first bar). We found that protein extraction could be increased on average by 1.5 fold when doubling the beads’ size (*p* = 0.12). We also found that protein extraction is significantly higher after two homogenization cycles (1.7-fold difference, *p* < 0.05). Furthermore, after a second homogenization cycle, the better performance of the larger beads becomes more evident (protein yield for medium-sized and larger beads: 1.4 ± 0.5% versus 2.2 ± 0.7% m protein/m tissue, *p* = 0.06, not shown in Figure 2). Hence, after these tests, we concluded that protein extraction is greater using 2.8 mm beads in two homogenization cycles.

### 3.3. The Effect of Tube Capacity, Further Homogenization Cycles, and Increasing Lysis Buffer Volume to Tissue Mass Ratios

Next, we wanted to test whether the O-ring cryotube capacity affects protein extraction and whether we could improve yield by extending the number of homogenization cycles and/or by increasing the relative amount of the lysis buffer. For these experiments, we used the same amount of tissue from a single sample (30 ± 2 mg) to exclude biological variability′s effects and of varying tissue amounts. No changes were made from the previous protocols, except we either (a) changed the tubes (from 2 to 7 mL), (b) increased the number of homogenization cycles (2, 3, or 4 cycles of 5000 rpm for 30 s, always with a 5 min interval on ice), or c) increased the volume of lysis buffer relative to tissue amount (10, 15, or 20 µL/mg). Figure 3 shows the effect of these variables on the protein yield. We found no difference in using 2 mL or 7 mL tubes. Thus, we stuck with the smaller tubes to reduce waste and costs. Although not reaching significance, there is a clear downward trend in protein yield when we perform more than two beating cycles. The protein yield can reduce by approximately 60% when samples are subjected to four cycles. We attribute this reduction to the excess kinetic energy generated during the process, which may result in the degradation of most labile proteins. Therefore, we recommend no more than two cycles and cool the samples down in ice for at least 5 min between those cycles.

On the other hand, we observed a significant improvement in protein yield when increasing the relative amount of lysis buffer. Indeed, by doubling the volume of lysis buffer, we obtained, on average, twice as much protein. The results suggest that increasing the lysis buffer over 20 µL/mg may result in better protein extraction, but this will always be limited by the recommended maximal volume filling of the tubes (1.4 mL) and the relative occupancy of beads (based on the previous experiments, we fixed the beads-to-tissue mass ratio at 40). Therefore, in these optimal conditions, we do not recommend processing more than 45 mg of tissue in 2 mL (900 µL lysis buffer + 432 µL of beads, whose density is ~3.7 g/mL). If larger amounts of tissue need to be processed, 7 mL tubes should be used.

### 3.4. The Effect of Lysis Buffer Chemistry

Apart from RIPA, strong urea buffers are commonly used as lysing buffers, especially when aiming to enrich phosphoproteins. Therefore, we wanted to compare these two buffers′ performance under the previously optimized conditions, except we used 10 µL lysis buffer/mg of tissue to obtain more concentrated extracts for a downstream SDS-PAGE analysis. Again, we used three biological replicates fairly divided by the different experiments (30 ± 1 mg). After the second homogenization cycle, the first supernatant was centrifuged as described previously, except we used a second 2 mL O-ring tube. After saving the first protein-rich supernatants, we added 0.4 g of 1.4 mm beads and 100 µL of the other alternative lysis buffer to further extract proteins from the pellet. This was accomplished by an extra homogenization cycle in the same settings (5000 rpm for 30 s). The resulting extract was cleared by spinning down for at least one minute until no foam was visible. The protein concentration of the supernatants from the first and second extraction was estimated (Table 1) and an SDS-PAGE profile was obtained for all samples (Figure 4).

Although a comparable protein yield and protein electrophoretic profile were attained when processing the samples with RIPA or urea buffers, on a second extraction, it became clear that RIPA buffer was superior, guaranteeing about twice as much protein from the first pellet. Moreover, a more diverse range of proteins seems to be obtained with RIPA buffer, as judged by the higher number of automatically detected bands after a second extraction (Figure 4, gel on the right). Therefore, unless the downstream protein analysis requires the use of the urea buffer, we recommend the use of RIPA buffer to extract proteins from the AV tissue.

Most of the previous studies on aortic valve proteomics have followed a conventional liquid nitrogen grinding strategy [5,6,7,11]. Less used strategies include sonication on ice [13] or mechanical disruption with upright homogenizers [8,9]. To the best of our knowledge, only Weisell et al. [10] have attempted a similar BBS in a different instrument (MagnaLyzer). However, since methodological optimization was not the scope of their work, no details were given on the protein yield, type or size of the beads, tube capacity, number and duration of the homogenization cycles, nor the lysis buffer/tissue mass ratio, precluding the reproduction of the findings. Herein, we sought to optimize a BBS method aiming to maximize protein yield. Following the results of all assays, we conclude that an optimal protein extraction is attained by homogenizing the AVs in 2 mL O-ring cryotubes with 2.8 mm zirconium oxide beads, in 2 bead-beating cycles, in the presence of 20 µL of RIPA buffer/mg tissue (Figure 5). Based on these conditions, we have designed a standard operating procedure (available in Appendix A for consultation). Depending on the downstream applications, it is still possible to use urea-based buffers with satisfactory results. Further, when aiming at larger amounts of concentrated proteins (important, for instance, for Western blot experiments), we can decrease the lysis buffer/tissue mass to 10 µL/mg, while increasing the amount of tissue. In the case of human samples, this is generally not a problem, as resected AVs usually weigh over 1 g.

### 3.5. Exploring the Sexual Dimorphism in Aortic Valve Stenosis through Proteomics: A Proof-of-Concept Demonstration of the Usefulness of the SOP for Aortic Valve Homogenization and Protein Extraction

Having optimized the protocol for AV homogenization and protein extraction, we then aimed to test its suitability for proteomics applications. To this end, we used the sexual dimorphism in AS as a case-study. For the same level of stenosis, women show higher levels of fibrosis and men present higher calcification [22]. These differences are reflected in different calcium score cutoffs to define severe AS by computed tomography: > 3000 for men and > 1600 Agatston units for women [23]. Thus, we followed a proteomics approach to determine whether there are differentially expressed proteins between men and women in AS, which may help explain the sexual differences in disease presentation. To this end, we used our SOP to homogenize the AVs and extract proteins from six AV samples (three men, three women, Table 2). The protein lysates were then subjected to SDS-PAGE separation, in-gel digestion, data-dependent acquisition analysis to generate ion libraries, and SWATH-MS analysis to perform relative protein quantification.

With the DDA approach, we could identify 1323 proteins or 1032 proteins if we only consider proteins identified with ≥2 unique peptides. From the 1323 proteins, we could successfully quantify 808 proteins across all 6 samples, or 747 if we only consider proteins that had been identified with ≥2 peptides (Appendix A). The proteins were quantified across a range of six orders of magnitude (Figure 6, left panel), which we compared to Aikawa’s group [6] (Figure 6, right panel), currently the largest AV proteome dataset available, including analysis of heterogeneous AV samples (normal, fibrotic, and calcific) from nine donors. The comparability is limited because, despite the use of the same lysis buffer type (RIPA buffer), different LC modalities (Aikawa’s group used a 300 µm × 150 mm column, set a flow rate of 300 nL/min, and a 120 min gradient, while we used a 75 µm × 250 mm column, set a flow rate of 5 µL/min and a 60 min gradient) and mass spectrometers (Triple TOF 6600 vs. Q Exactive) were used. However, the spectrometers offer a similar linear dynamic range (>5 units), and we found our dataset strongly correlated with that of Aikawa’s group [6] (Spearman coefficient 0.71, *p* < 10^−16^, Appendix A). In the latter case, and using the same quality criteria (proteins identified with >2 peptides) and the same principles of SWATH-MS quantification, to control for bias in the comparison, we estimated a dynamic range of 4.4 orders of magnitude. Therefore, at the very least, our results suggest a non-inferior performance of our SOP, using a BBS, compared to standard liquid nitrogen pulverization used in [6], for AV proteome characterization. Of note, our work is limited to the analysis of the canonical protein isoforms. In the future, the performance of both homogenization methods should be compared with regard to the depth of proteoforms profiling.

As for the specific biological question, we found 16 putative proteins dysregulated by sex, including 12 upregulated proteins in women and 4 upregulated proteins in men (Figure 7 and Appendix A). Curiously, matrix extracellular proteins such as periostin, vinculin, collagen alpha-1(XVIII) chain and leucine-rich alpha-2-glycoprotein, and focal adhesion proteins, such as integrins alpha-V and beta-1, which are responsible for establishing a tight cellular-extracellular matrix contact and for mechanotransduction, were found increased in women. In men, we found higher levels of specific immunoglobulins and of the membrane-associated phospholipase A2. These results, of course, demand confirmation in an adequately powered study. However, for now, we can say that the results are in line with previous observations of sexual dimorphism in AS. For instance, the increased expression of extracellular matrix or focal adhesion proteins is expected in more fibrotic tissues, as happens in female stenotic valves (refer to Figure 8, a protein–protein interaction analysis of the dysregulated proteins). In turn, the relative increase in phospholipase A2 in men, which often present exuberant AV calcification, corroborates previous studies showing the pro-calcific role of lipoprotein-associated phospholipase A2 in valve interstitial cells [24], or the association between phospholipase A2 activity (in lipoproteins) and coronary calcification (a cardiovascular condition with similar etiology) in the Rotterdam Coronary Calcification Study [25].

## 4. Conclusions

We have developed an SOP for an optimal and more reproducible homogenization of AVs and protein extraction. Protein extraction from calcified AVs was optimal using 2 mL O-ring tubes filled with 2.8 mm zirconium dioxide beads (40× the tissue mass), with no more than two homogenization cycles, in the presence of RIPA buffer. The higher the RIPA buffer/tissue mass ratio, the greater the protein yield, but the lower the concentration of the protein lysates. This protocol was successfully applied to the characterization of the AV proteome in patients with AS, covering a dynamic range of six orders and showing the potential to dissect the sexual dimorphism in this condition at the protein level.

## Figures and Tables

**Figure 1 proteomes-10-00030-f001:**
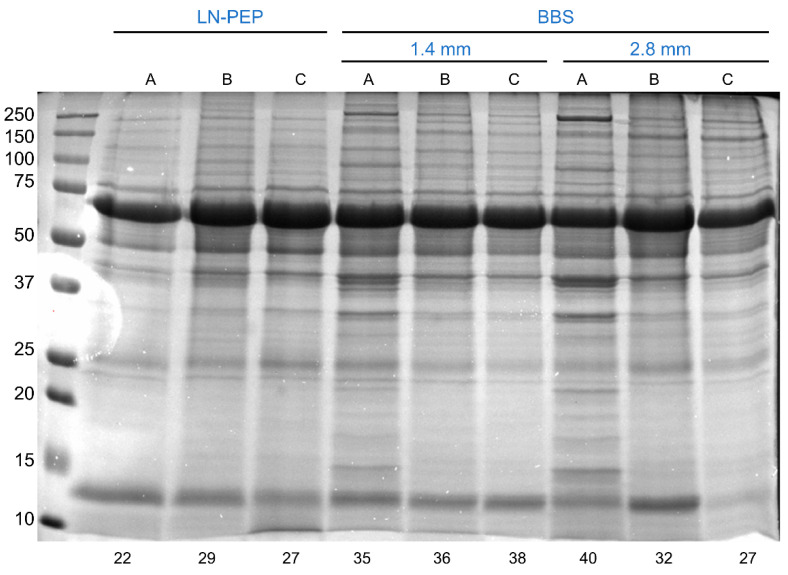
SDS-PAGE profile of the aortic valve lysates obtained after liquid nitrogen grinding + mechanical disruption with a Potter system (LN-PEP) or by homogenization with a bead-beating system (BBS). A, B, and C refer to the distinct aortic valve samples. The number of automatically detected bands is given below each lane.

**Figure 2 proteomes-10-00030-f002:**
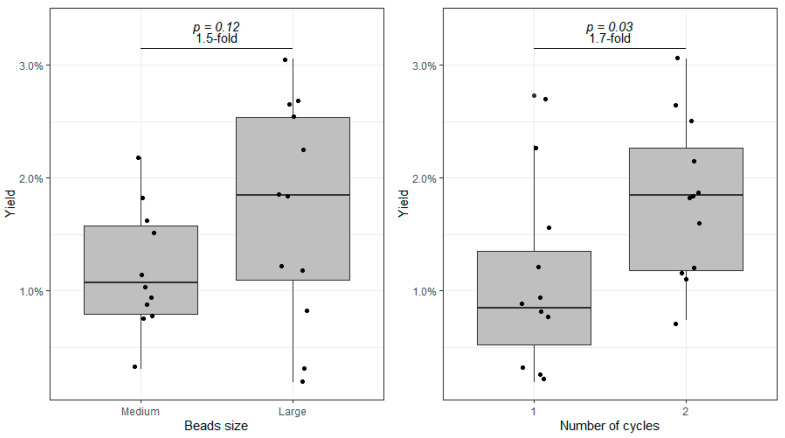
Comparison of the protein yield (protein mass/tissue mass) between medium (1.4 mm) and large (2.8 mm) zirconium dioxide beads (**left**), and between one versus two homogenization cycles (**right**). Boxes represent the median and interquartile range.

**Figure 3 proteomes-10-00030-f003:**
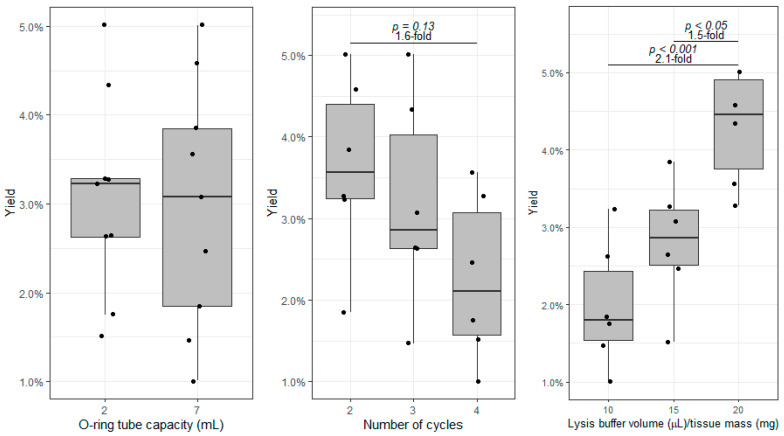
Comparison of the protein yield (protein mass/tissue mass) between 2 and 7 mL O-ring tubes (**left**), between two, three, and four homogenization cycles (**middle**), and between increasing lysis buffer volume/tissue mass (µL/mg) ratios (**right**). Boxes represent the median and interquartile range.

**Figure 4 proteomes-10-00030-f004:**
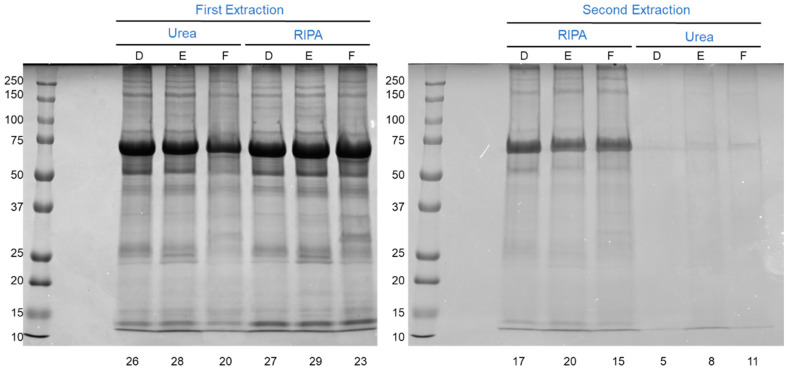
SDS-PAGE profile of the aortic valve lysates obtained after a first extraction with urea or RIPA buffers (**left**) and after a second extraction with RIPA or urea buffers. Twenty-five micrograms of protein were loaded in the gel on the left and 20 µL of the second extract (protein much more diluted) was loaded in the gel on the right. D, E, and F refer to distinct aortic valve samples. The number of automatically detected bands is shown below each lane.

**Figure 5 proteomes-10-00030-f005:**
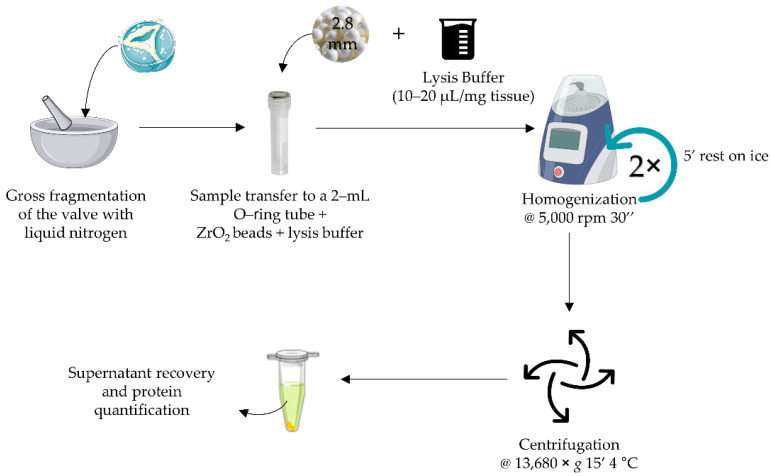
Overview of the workflow for the homogenization of the aortic valves and protein extraction, according to the SOP.

**Figure 6 proteomes-10-00030-f006:**
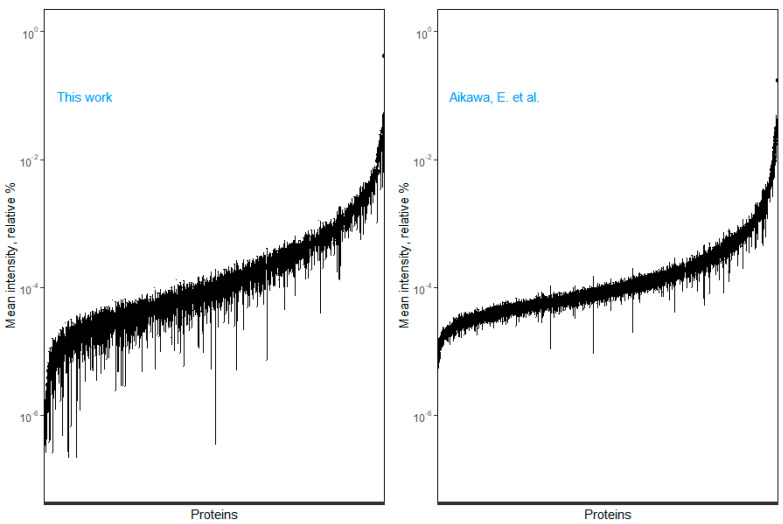
Proteome dynamic range, based on relative quantification. The 95% confidence interval of the mean % of each protein is depicted from the least to the most abundant protein (albumin). Our proteome dataset is presented on the left. On the right, we applied the same principle of SWATH-MS analysis into the Aikawa et al. dataset, the largest aortic valve proteome dataset published to date ([6]). The width of the confidence intervals reflects the number of samples analyzed (6 vs. 27). Protein names are not shown due to space constraints. In any case, we only selected proteins identified with at least two peptides.

**Figure 7 proteomes-10-00030-f007:**
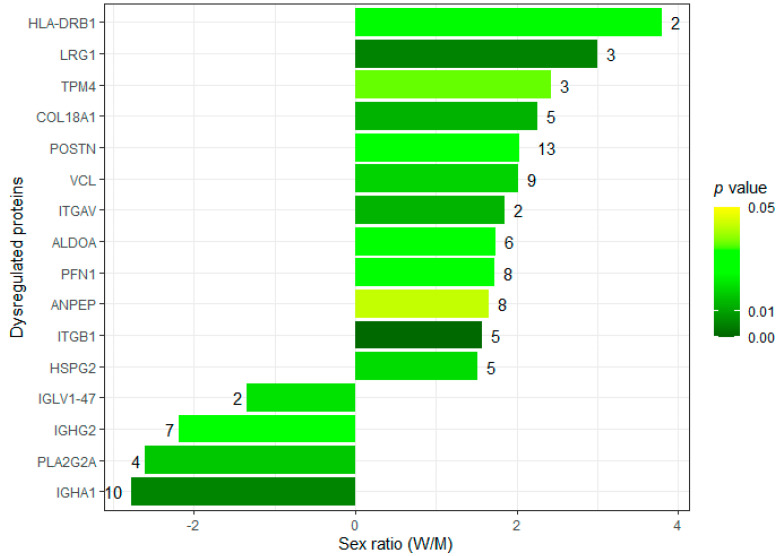
Proteins dysregulated between men and women, according to the SWATH-MS analysis of six aortic valves. Proteins are identified with the respective gene name. The number of peptides used for quantification is given on top of the bars. The darker green bars mark the proteins showing the most significant differences.

**Figure 8 proteomes-10-00030-f008:**
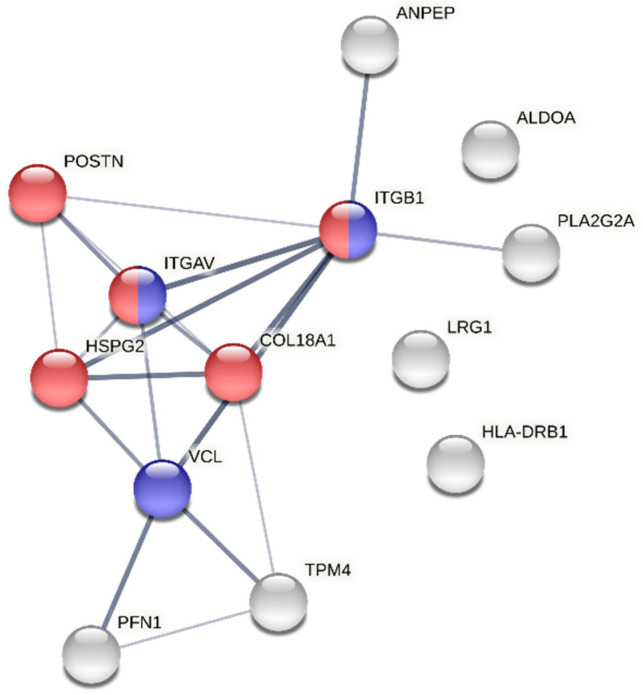
Protein–protein interaction network of the aortic valve proteins dysregulated by sex. The edge thickness is proportional to the confidence score of the interactions. Proteins involved in the extracellular matrix organization (GO:0030198) are marked in red, and proteins participating in cell–matrix adhesions (GO:0007160) are marked in blue. In any case, these proteins were increased in women’s valves. The protein PLA2G2A (phospholipase A2, membrane-associated) was the only protein upregulated in men mapped by STRING.

**Table 1 proteomes-10-00030-t001:** Comparison of the Protein Extraction Performance between RIPA and Urea Lysis Buffers.

**First Lysis Buffer**	**Protein Yield (mg)**	***p*-Value ^1^**	**Protein Yield (%)**	***p*-Value ^1^**
RIPA	0.55 ± 0.08	0.88	1.8 ± 0.2	0.74
Urea	0.57 ± 0.17		1.9 ± 0.6	
**Second Lysis Buffer**	**Protein Yield (mg)**	***p*-Value ^1^**	**Protein Yield (%)**	***p*-Value ^1^**
Urea	0.06 ± 0.05	0.09	0.2 ± 0.2	0.07
RIPA	0.14 ± 0.02		0.5 ± 0.1	

^1^ Unpaired two-tailed *t*-test.

**Table 2 proteomes-10-00030-t002:** Characteristics of Aortic Stenosis Patients.

	Men (n = 3)	Women (n = 3)
Age, years	70 ± 8.2	72 ± 8.1
Diabetes (type 2), n	1	2
Smoking (active or past history), n	1	0
Arterial hypertension, n	3	3
Dislipidemia, n	3	3
Body Mass Index, kg/m^2^	29.1 ± 4.9	32.0 ± 4.5
Indexed aortic valve area, cm^2^/m^2^	0.5 ± 0.07	0.4 ± 0.05
Mean transvalvular pressure gradient, mmHg	54 ± 11	53 ± 16
Indexed left ventricle mass, g/m^2^	147 ± 48	132 ± 28

Continuous data are presented as the mean ± SD. No significant differences were evident.

## Data Availability

The MS data have been deposited to the ProteomeXchange Consortium via the PRIDE partner repository with the dataset identifier PXD034386.

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
