# Peer review of "Optimization of a Protocol for Protein Extraction from Calcified Aortic Valves for Proteomics Applications: Development of a Standard Operating Procedure"

_proteomes, 2022, doi:10.3390/proteomes10030030_

Round 1

Reviewer 1 Report

1) Please add a workflow diagram for sample preparation which would make it much easier to understand. 

2) Figure 1 say bead size as 1.4mm and 1.8mm whereas line 248 mentions bead size as 2.8mm. Please correct appropriately. 

3) What was the amount of tissue samples taken to begin with LN-PEP & BBS and what was the final protein yield in mg.

4) Figure 4: Crop and re-arrange the figure to maintain consistency between 1st and 2nd extractions gels. 

5) In table 1, also provide protein yield in mg. 

6) Section 3.4 - It is confusing how 2nd extraction was done, do authors mean 2nd homogenization cycle to the same leftover pellet from the first extraction. If yes then 1) I am not sure if the conclusion that RIPA is superior to urea extraction buffer because 1) urea extracted most of the proteins within the first cycle thus reducing the additional 2nd cycle need 2) Although RIPA shows multiple bands in extraction 2 authors have not mentioned if these same bands were present in extraction 1 or if there are any additional bands identified, if there are no additional bands then both extraction methods have more or less similar number of bands. 

7) Ideal experiment would be to determine number of proteins identified and their sequence coverage after each extraction from the same sample with different extraction methods than just relying on protein yield.  

Reviewer 2 Report

bead beating of calcified aortic valves for

Reviewer 3 Report

It is recommended to reconsider whether to publish this work if the following comment can be responded, and suggestions can be addressed in revised manuscript.

1)      In line 212, why the author initially used RIPA buffer to extract proteins rather than urea buffer which may provide more protein yielding than RIPA?

2)      In figure 2a, please remove the bottom line in x axis name “Beads_size”.

3)      In line 250, if the p = 0.12, it may be hard to claim beads size really matters in protein yielding.

4)      Did the author use other protein quantification method in addition to protein DC assay? E.g. BCA and short-gel. It may not be super accurate to use one assay to measure the protein yielding from different system.

5)      When comparing the RIPA and Urea lysis buffer, why the author didn’t combine the two fractions of 1st and 2nd extraction from each buffer and then compare the lysate by SDS-PAGE?

6)      It would be better to label the left panel in figure 5 as “this work” since this figure aim to compare two works.

7)      Beside the ranked protein plot, did the author use other correlation method to compare the consistency between two works? E.g. person or spearman correlation analysis.

8)      Have the author perform PCA analysis to see if the samples can be clustered by sex?

9)      In line 372, how up-regulation and down-regulation were defined? By fold change or p value or both? Is the author able to show the overall protein relative quantification heatmap or volcano plot?

10)   In figure 6, how sex ratio was calculated? Does the author use fold change or something else?

11)   In figure 6, please remove the bottom line in legend name “p_val”.

12)   The author mentioned DIA acquisition in method part, but I didn’t see any result from DIA.

Round 2

Reviewer 3 Report

The author of manuscript have responded to all comment and revised the manuscript according to my suggestions. It is recommended to publish this work without any further revision.